# Building Resilience and Social–Emotional Competencies in Elementary School Students through a Short-Term Intervention Program Based on the SEE Learning Curriculum

**DOI:** 10.3390/bs14060458

**Published:** 2024-05-29

**Authors:** Hee Jung Min, Sang-Hee Park, Seung-Hyun Lee, Bo-Hwa Lee, Mikyung Kang, Mi Ju Kwon, Myung Ju Chang, Lobsang Tenzin Negi, Tsondue Samphel, Seunghee Won

**Affiliations:** 1Department of Children and Adolescent Education, Dongguk University-WISE, Gyeongju 38066, Republic of Korea; aravindamhj@gmail.com; 2Daegu Student Suicide Prevention Center, Kyungpook National University Chilgok Hospital, Daegu 41404, Republic of Korea; bomhangi@hanmail.net (S.-H.P.); pigmandarin@gmail.com (S.-H.L.); ibohwa030@gmail.com (B.-H.L.); godk-mk@hanmail.net (M.K.); pkwkmj@naver.com (M.J.K.); 2-yab@hanmail.net (M.J.C.); 3Center for Contemplative Science and Compassion-Based Ethics, Emory University, Atlanta, GA 30322, USA; snegi@emory.edu (L.T.N.); tsamphe@emory.edu (T.S.); 4Department of Psychiatry, School of Medicine, Kyungpook National University, Daegu 41944, Republic of Korea

**Keywords:** COVID-19, elementary students, resilience, social–emotional development, SEE learning

## Abstract

This study explored the positive effects of a six-week Social–Emotional and Ethical Learning^®^ (SEE Learning) program on resilience and social and emotional competences, adapted for elementary students in Daegu, South Korea, a region strongly affected by the first outbreak of COVID-19. A total of 348 third- and fourth-grade students from 15 elementary schools participated, and the curriculum was tailored, emphasizing key areas such as resilience, attention, kindness, attention training, and compassion. Repeated measures analysis of variance (RMANOVA) tests showed statistically significant improvements between pre- and post-tests in resilience and its subscales, including self-efficacy, tolerance of negative affect, positive support relations, power of control, and spontaneity, as well as in social and emotional competencies, including emotional regulation, social skills, empathy, and social tendencies. Despite a lack of maintenance in all areas, at follow-up, the mean scores for self-efficacy, tolerance of negative affect, and positive support relations, as well as emotional regulation, social skills, empathy, and social tendency, remained higher than pre-test levels, suggesting some lasting benefits. The findings underscore the potential of the SEE Learning program integrated with resilience, mindfulness, compassion, and ethical practices to enhance students’ resilience and social and emotional well-being. This study contributes to the growing body of evidence supporting the use of mindfulness and compassion-based SEL programs to mitigate the adverse effects of traumatic events on children’s mental health.

## 1. Introduction

The onset of remote learning and the absence of normal school life due to the emergence of the COVID-19 pandemic deprived children of opportunities and venues to engage in social relationships, leading to a lack of time for self-reflection and forming connections with others [1]. Consequently, students have experienced psychological issues such as depression and anxiety, as well as fear related to academics and peer relationships [2]. Elementary schools play a critical role as educational institutions that foster fundamental knowledge, healthy peer relationships, and balanced social, emotional, cognitive, and physical development, thus contributing to the upbringing of a “whole child” [3]. The pandemic disproportionately impacted elementary students who began their schooling around 2020, as they were unable to engage in typical school life and form adequate peer relationships, omissions that exacerbated their psychological and relational difficulties [4].

Several studies have shown that students have experienced negative consequences as a result of the COVID-19 pandemic. For example, in an examination of the mental health of adolescents in Daegu, South Korea, which was the site of the earliest COVID-19 outbreak in South Korea and thus had experienced very strict quarantines [5], working with the Daegu Student Suicide Prevention Center at Kyungpook National University Chilgok Hospital, found increased levels of stress, anxiety, and depression among students in 2020 [6]. According to a 2021 survey by the Korean Statistical Classification and the Ministry of Gender Equality and Family of Korea, 48.4% of students reported negative changes in their school life [7]. Additionally, an analysis of longitudinal data from 15,000 students across elementary, middle, and high schools in Seoul Metropolitan City, conducted by the Seoul Education Research and Information Institute, revealed a statistically significant increase in depression, worry, and irritability among elementary students in 2020 compared to 2019 [8]. These types of negative experiences in childhood can lead to various mental disorders or psychological difficulties in adulthood, highlighting the need for interventions aimed at helping students overcome these challenges and supporting their healthy development [9].

In 2022, with elementary students experiencing low levels of school engagement and high levels of depression and anxiety, South Korean government mandated the resumption of face-to-face classes, which are essential for school attendance and peer interaction. However, students and teachers were unprepared for the return to in-person classes; students struggled with emotional regulation and increased impulsivity, resulting in maladjustment to school life and creating challenges for both students and teachers [10]. Consequently, in alignment with global trends, Korea has an intensified need for social and emotional learning (SEL) to alleviate the negative feelings experienced during COVID-19 and improve mental health, thereby fostering students’ emotional and relational well-being [11]. Kim and Shin [12] discovered that children’s social and emotional competencies positively affected peer relationships and enhanced school adaptation during the COVID-19 pandemic, a finding that underscores the importance of developing children’s social and emotional skills. While researchers and educators have recognized the need to enhance socio-emotional competencies in the past, the COVID-19 pandemic led to a renewed emphasis, resulting in a stronger call for the introduction of socio-emotional programs in the educational sector [13]. Therefore, applying and demonstrating the effectiveness of social and emotional learning programs have become essential tasks, especially in Daegu, the first city in South Korea to face a COVID-19 outbreak.

SEL began in the United States as a preventive aspect of character education aimed at promoting social development by resolving the conflicts that arise in relationships. Educators began to recognize that SEL programs, which were initiated as a research project at Yale University in the 1960s, yielded positive changes in behavior, emotions, and academic achievement. In 1994, the Collaborative to Advance Social and Emotional Learning (CASEL) was formed, and in 1995, Daniel Goleman published his book *Emotional Intelligence*. These developments prompted the spread of SEL programs across the United States and globally. In 2001, CASEL changed its name to the Collaborative for Academic, Social, and Emotional Learning; the group actively promotes the introduction of SEL programs in schools. CASEL defines the five key components of SEL as self-awareness, self-management, social awareness, relationship skills, and responsible decision-making, and the group aims to cultivate these elements to help students respect themselves and others, establish healthy relationships, and solve problems constructively to lead healthy lives [14]. Various other SEL programs, combined with mindfulness and compassion curricula, are being used to help individuals find meaning in life and become happy by living proactively and constructively [15,16].

The Center for Contemplative Science and Compassion-Based Ethics at Emory University, in collaboration with the founders of CASEL programs, has developed a new social–emotional educational model, SEE Learning (Social, Emotional, and Ethical Learning). SEE Learning incorporates trauma-informed education, resilience, attention training, ethical discernment, compassion, and systems thinking into the existing SEL framework, thereby establishing an extensive new line of social–emotional education. SEE Learning entails a new social–emotional educational model that emphasizes sustainable and self-motivated learning, and differs from traditional SEL in its integration of attention training and compassion. Since its release in 2019, SEE Learning has been applied in multiple Korean schools; however, no research has been conducted on its outcomes. Therefore, there is a need to explore the effectiveness of the SEE Learning Curriculum—especially its impact on the social–emotional development of elementary students who have returned to face-to-face classes for the first time post-COVID-19—in Korean classrooms.

This study aims to develop and implement a short-term intervention program to apply the SEE Learning program in a post-COVID-19 face-to-face setting and to explore its effectiveness. The necessity of developing a short-term intervention arises in situations such as those brought about by COVID-19, where normal curriculum progress is challenging, when the entire curriculum is too lengthy to be incorporated into the public standardized educational system, or when there is a lack of understanding of the curriculum. Therefore, in this post-COVID-19 situation and under circumstances where a year-long curriculum cannot be implemented, we aimed to develop and apply a short-term intervention program of the SEE Learning program and investigate its effectiveness.

To achieve the purpose of the study, we implemented a short-term intervention based on the SEE Learning program, which integrates resilience, attention training, and compassion, in Daegu area elementary schools where face-to-face classes had resumed following the initial COVID-19 outbreaks. We then explored changes in students’ social and emotional development, including resilience, emotional regulation, social skills, and empathy. In this article, we outline the SEE Learning program utilized in the study and discuss the relevance and development of social and emotional learning in the context of COVID-19. Additionally, we examine the emotional and relational shifts students experienced after program implementation and the broader implications of these shifts.

## 2. Literature Review and Research Hypothesis

### 2.1. Social–Emotional Learning (SEL)

Social and emotional education refers to education and training focused on social and emotional skills, attitudes, and behaviors that are crucial for human development; this type of education is broadly referred to as social–emotional learning (SEL). SEL encompasses abilities such as being aware of one’s own emotions, labeling these emotions, and understanding, accepting, and ultimately regulating them. These competencies form the foundation for SEL, which also trains individuals in competencies necessary for understanding others’ emotions and effectively forming and maintaining relationships [14].

Over time, this approach was adopted widely as educators recognized that it not only improved social and emotional competencies but also had a positive impact on academic performance. The 1980s saw the establishment of various programs focused on social and emotional competencies, including the “New Haven Social Development” program [17]. Then, in 1994, scholars from a range of fields (including Daniel Goleman, Mark T. Greenberg, Eileen R. Growald, Linda Lantieri, Timothy P. Shriver, David J. Sluyter, and Roger P. Weissberg), having recognized a “missing piece” in the educational field, came together to establish CASEL; since that time, the term “social–emotional learning” has been used to describe this aspect of education [14]. The global spread of SEL was bolstered in 1995 when Daniel Goleman, a co-founder of CASEL, published his book *Emotional Intelligence*, sparking worldwide interest in emotional development [18]. Interest continued to grow following the publication of Goleman’s work on social intelligence, prompting further focus and research on social and emotional competencies. Importantly, basic social and emotional competencies have likely been cultivated among teachers, students, and families even before these movements. However, the formalization and programmatic delivery of these skills have led to increased research, highlighting their necessity and importance. Reflecting these developments, CASEL now defines SEL as the development by both children and adults of the knowledge, skills, and attitudes needed to establish a healthy identity, regulate emotions, achieve personal and collective goals, form empathetic and supportive relationships, and make responsible and caring decisions.

CASEL has defined five key components that form the most widely used theoretical framework within the SEL context: self-awareness, self-management, social awareness, relationship skills, and responsible decision-making. Briefly, self-awareness refers to the ability to identify one’s own sensations, emotions, and thoughts, as well as being aware of one’s actions. Self-management is the ability to regulate one’s emotions, thoughts, and behaviors. Social awareness involves the ability to recognize and empathize with the emotions, thoughts, and behaviors of others. Relationship skills denote the capacity to initiate and maintain healthy relationships. Lastly, responsible decision-making is the ability to make moral and rational decisions within relationships [19,20]. In 2018, CASEL expanded the original framework by incorporating the concept of equity, detailing SEL competencies that encompass culturally responsive and relevant teaching, community building, and efforts to develop students’ ethnic and racial identities alongside project-based and experiential learning opportunities. CASEL continues to lead the SEL movement as one of the most prominent SEL associations, offering clear, evidence-based guidance.

In 2019, the Center for Contemplative Science and Compassion-Based Ethics at Emory University introduced an expanded version of SEL programs known as SEE Learning [21]. SEE Learning was created by adding an emphasis on the ethical aspects that were seen as missing in existing SEL programs; specifically, the revised approach incorporates attention training, compassion, and ethical practices. Further, SEE Learning is based on the assumption that all students may carry traumas, big or small, and thus, the program integrates skills that help students effectively overcome and manage everyday traumas, thereby fostering resilience and offering trauma-sensitive education. In sum, SEE Learning represents a new approach to social–emotional learning that surpasses the basic concepts and methodologies of previous SEL.

Research on the effects of SEL programs has produced varied results. Durlak et al. [22] reviewed 12 meta-analyses of SEL program research and found that SEL programs have statistically significantly enhanced the social skills, prosocial behaviors, and academic achievement of one million students and have statistically significantly reduced emotional distress and conduct problems. Other studies have shown positive effects on students’ resilience [23,24,25] and school functioning or academic performance [26,27]. In South Korea, the continued publication of studies applying SEL and verifying its educational effectiveness shows its growing importance. Park and Chae [28] conducted a meta-analysis of 20 papers (published from 2011 to 2020) on the effects of SEL programs, which revealed an effective increase in social and emotional competencies. Additionally, researchers have conducted studies aimed at enhancing the effectiveness of SEL via the integration of mindfulness [15], and the Seoul Education Research Information Institute has discussed integrating SEL with mindfulness or moral education in the elementary school curriculum [16,29]. Further, Hong et al. [30] reported increases in empathy and self-regulation skills due to SEL programs. Various studies have suggested that mindfulness and/or compassion-based programs can improve resilience, self-regulation, attention, empathy, social connectedness, and positive feelings and can decrease stress, depression, and anxiety [31,32,33,34,35,36,37,38,39].

Overall, prior research on SEL programs suggests that these programs enhance empathy, emotional regulation, and social skills. However, there is a notable gap in the research on the effects of recently improved programs, such as the mindfulness and compassion-based SEE Learning program, particularly their impact on resilience as well as social–emotional competencies. Thus, there is a need for further research in this area.

### 2.2. Social–Emotional and Ethical (SEE) Learning

Launched globally in 2019, the SEE Learning program provides a theoretical framework for compassion-based ethics and is applicable to both K-12 and higher education as a social, emotional, and ethical learning curriculum. Developed by the Center for Contemplative Science and Compassion-Based Ethics (CCSCBE) at Emory University, SEE Learning aims to educate both the heart and mind. Building on the foundation of existing SEL programs, SEE Learning incorporates the social and emotional competencies addressed in conventional SEL programs and expands to cover aspects previously missing from traditional SEL frameworks. These additions include attention training based on mindfulness, awareness, and heedfulness; compassion and kindness activities; ethical discernment; resilience and trauma-informed practices; systems thinking; equity; and sustainability.

SEE Learning is based on secular ethics and addresses common ethical principles that apply to everyone, regardless of their beliefs or faiths. Further, it provides a framework based on the three core skill sets in the classroom, as suggested by Goleman and Senge [40]. These skill sets extend training to the self, others, and larger systems. Additionally, scholars researching mindfulness, education, compassion, psychology, trauma, resilience, and SEL have collaborated to create the theoretical model underlying the SEE Learning Curriculum. This model encompasses three dimensions—awareness, compassion, and engagement—and operates within three domains: personal, social, and systems. This 3 by 3 framework contains nine fundamental components: attention & self-awareness, self-compassion, self-regulation, interpersonal awareness, compassion for others, relationship skills, appreciating interdependence, recognizing common humanity, and community and global engagement. Consequently, SEE Learning expands upon conventional SEL, aiming to cultivate a more diverse and healthy heart and mind, thus ensuring comprehensive social, emotional, and ethical training.

This theoretical framework aims to foster a compassionate classroom environment as part of the curriculum, helping students to improve their resilience, awareness, and attention, navigate emotions, and learn about and understand others. Further, this curriculum emphasizes the interconnectedness of individuals in a system, highlighting how they can influence others and be influenced in return, underscoring their connectedness.

Specifically, the SEE Learning curriculum includes 43 learning experiences spread across nine chapters. These chapters cover topics such as creating a compassionate classroom, building resilience, strengthening attention and self-awareness, navigating emotions, learning about and from one another, compassion for self and others, we are all in this together, and building a better world. For examples, students learn, experience, and practice kindness by reflecting on the acts of kindness they have received through drawing activities. They explore diverse resilience strategies using ‘Help! Now!’ strategies and learn about the importance of attention and how it benefits them by cultivating mindfulness, awareness, and heedfulness. The curriculum also addresses understanding the needs behind various emotions and mechanisms to manage them, handling impulsive behaviors, and gaining insights into others’ lives through mindful interviews. Students recognize commonalities and differences, cultivate kindness and compassion for themselves and others, and understand systems within their classrooms, schools, and neighborhoods. They also appreciate interdependence through activities that help think about people related to their favorite food, achievements, and other vital aspects, and apply these insights within their communities.

Therefore, the SEE Learning curriculum can be described as an integrated and comprehensive program that enables students of various ages to develop the knowledge, skills, and attitudes necessary for a healthy life over the course of a year. The education provided is neither coercive nor strictly teacher-oriented; instead, students receive knowledge and engage in thoughtful reflection through various activities, progressing through stages of embodied understanding as they practice and deeply internalize the knowledge and skills. Thus, SEE Learning emphasizes critical thinking, thus facilitating proactive and autonomous learning [21].

### 2.3. Research Question

Since its release in 2019, SEE Learning has been implemented in approximately 50 countries worldwide. However, few preliminary empirical studies have been conducted on the outcomes and thus, there is a need for further validation of the SEE Learning curriculum’s effectiveness. To address this need, this study implemented the SEE Learning program in elementary school classes in the Daegu area, which was the site of the first COVID-19 outbreak in South Korea and experienced significant disruptions in schooling. The analysis examines changes among elementary students after returning to face-to-face classes once COVID-19 restrictions were lifted; specifically, we test whether changes in resilience, emotional regulation, social skills, and empathy accompanied the implementation of the SEE Learning program, which integrates mindfulness, compassion, resilience, and ethical approaches [41].

## 3. Research Methods

### 3.1. Study Design

The study was conducted from February 2022 to March 2023 with the approval and support of the Daegu Metropolitan Office of Education. We obtained consent from classroom teachers, parents, and students. We provided participating classroom teachers with two hours of orientation training and sent parents invitation letters detailing the program and its contents. Students also received an overview of the research project. All procedures for the study were approved by the institutional review boards of both the university hospital and school district (Reg. No. KNUCH 2023-06-028).

### 3.2. Participants

Schools volunteered to participate in the study in response to research flyers. Over 30 schools applied, and we selected 15 classrooms in 15 elementary schools spread across public school districts in Daegu Metropolitan City. The SEE Learning program for students in grades 3–4 was adapted for use in these schools. A total of 348 students from these classrooms participated, including 215 boys (61.8%) and 133 girls (38.2%), with 140 students in grade 3 (40%) and 208 in grade 4 (60%). Individual assessments were conducted before and after the six-week SEE Learning program. Among the participants, 156 boys and 79 girls, totaling 235 students (66.38% boys, 33.62% girls), completed a follow-up test three to four months later (a period that included school vacations). Student participants received a modified six-week SEE Learning program tailored to be developmentally appropriate for grades 3–4. Students with disabilities were allowed to participate and were not excluded from the study; however, there were no students with physical, developmental, or intellectual disabilities that would seriously interfere with the program proceedings.

### 3.3. Short-Term Intervention Program Based on the SEE Learning Curriculum

#### 3.3.1. Program Development Process

To implement the SEE Learning program in public elementary schools with students facing negative experiences and traumas associated with COVID-19, it was necessary to modify the curriculum to fit into a short six-week period in public schools to enhance resilience and social and emotional competencies. Thus, we reconfigured the program to incorporate 12 learning experiences over six weeks, focusing on aspects such as attention and self-awareness, self-compassion, self-regulation, interpersonal awareness, compassion for others, relationship skills, and appreciating interdependence. During this process, the research team, which included a psychiatrist, a psychiatric nurse, an education scholar specializing in SEE Learning, and six SEE Learning instructors designated to deliver the program, collaboratively reviewed and discussed the program activities.

#### 3.3.2. Program Structure and Content

Overall, the six-week, 12-session program based on SEE Learning focused on improving resilience and social and emotional competencies among elementary students. In the first week, students engaged in activities including self-introduction with an “in and out” activity, exploring kindness, and remembering kindness received. In the second week, they established classroom rules and practiced kindness. In the third week, they explored authentic kindness, conducted interdependence activities, and considered kindness connected to their achievements. In the fourth week, they contemplated body sensations, discovered new things through attention activities, practiced “Help! Now!” strategies, and learned how to resource. In the fifth week, they created a happy treasure box, practiced grounding techniques, and learned how to stay in the resilient zone. In the final week, they delved deeply into the resilient zone, learned about the nervous system, and explored nurturing mindfulness, awareness, and compassion together, using the case of understanding how to walk a dog. This short-term intervention program allowed students to discuss and actively engage in activities designed to improve resilience and social–emotional skills. Instructors facilitated these activities, and students actively engaged in each lesson by moving, drawing, acting, and discussing the subjects.

An education session suitable for elementary students generally lasts 30–40 min. In this study, to align with the school district’s schedule, each session was extended to 90 min, comprising two 40 min sessions of instruction with a 10 min break in the middle. Over the six-week period, there were 12 sessions, totaling 9 h of training activities (including break times). Each session began with a check-in to acknowledge individual feelings and included a 1 min mindfulness and compassion meditation that offered students tools for self-regulation in difficult times and used a trauma-sensitive approach. At the end of each session, instructors conducted a debriefing and another 1 min mindfulness meditation, creating space for students to share their thoughts and feelings.

#### 3.3.3. Program Application Process

Six experienced instructors who had completed official teacher trainings for the SEE Learning curriculum learned, and practiced each learning activity of the SEE Learning curriculum and had prior experience educating and counseling elementary students, led the program in schools. They delivered sessions in classrooms, integrating them into the regular school schedule. Classroom teachers were present during the sessions and participated in the program so that they could maintain and apply the curriculum after the SEE Learning instructors left. To ensure consistency in teaching, the instructors met as a group six times before the initiation of the program to review and revise the session plans, ensuring that their questions aligned with the objectives of the curriculum while maintaining uniformity.

### 3.4. Measures

#### 3.4.1. Resilience Scale for Children (RSC)

We measured students’ resilience via the Resilience Scale for Children (RSC) developed by Ju and Lee [42] for fourth-grade elementary students in South Korea. The RSC measure (overall α = 0.96) includes five subscales: self-efficacy (eight items, e.g., “I can do my best in the tasks given to me”, α = 0.90), tolerance of negative affect (eight items, e.g., “I can control unpleasant feelings”, α = 0.86), positive support relations (four items, e.g., “I can find someone to help me when needed”, α = 0.77), power of control (six items, e.g., “I can keep focusing on what I am interested in, even when interrupted”, α = 0.83), and spontaneity (four items, e.g., “I like to try new things”, α = 0.76). Items were rated on a five-point Likert-type scale, with higher scores indicating greater resilience.

#### 3.4.2. Social and Emotional Competencies Scale

To measure students’ social and emotional competencies, we used the scale of social–emotional competence integrated by Kim [43] and further revised by Choi [44]. The measure (overall α = 0.95) consists of five subscales: emotional regulation (13 items, e.g., “I don’t get angry and I accept losing when I lose a game”, α = 0.82), social skills (13 items, e.g., “I am the first to suggest what game to play when I am with friends”, α = 0.87), empathy (6 items, e.g., “I feel upset when I see a friend being bullied by another child”, α = 0.77), social tendency (8 items, e.g., “I am accepting and like peers or adults”, α = 0.76), and interpersonal relations (7 items, e.g., “I know how to express gratitude when I receive help from a friend”, α = 0.83). Items were rated on a five-point Likert-type scale, with higher scores indicating greater emotional regulation, social skills, empathy, social tendency, and interpersonal skills.

The questionnaires were standardized to a level understandable for grade 3–4 students. Surveys were collected directly by program instructors in collaboration with each classroom’s teacher. Questions were read aloud one by one, and students answered them by themselves. Additional explanations were provided only for questions that students did not understand. A 5 min break was taken once or twice, depending on the students’ conditions, and students who did not want to participate or wished to stop were allowed to do so. It took about 40–50 min to complete the survey, and beverages were provided during the process. After completing the questionnaires, the students’ conditions were checked.

### 3.5. Data Analysis

We employed Student *t*-tests to compare pre- and post-test scores for resilience and social–emotional competencies. For participants who completed the follow-up test, we conducted repeated measures analysis of variance (RMANOVA) to compare pre-, post-, and follow-up test scores to observe differences in resilience and social–emotional competencies over time and evaluate the consistency of the program effects. We then performed Tukey’s test to further examine the differences between the pre-, post-, and follow-up scores [45,46]. Students who transferred into or out of the class or were absent were excluded from the analysis as were cases with missing responses. Statistical significance was set to *p* < 0.05.

## 4. Results

### 4.1. Resilience

Table 1 presents the pre-test and post-test means and standard deviations for resilience and its subscales (self-efficacy, tolerance of negative affect, positive support relations, power of control, and spontaneity). A *t*-test comparing the mean scores before and after the SEE Learning intervention revealed statistically significant improvements in overall resilience (*t* = −3.82, *p* < 0.001) and all subscales: self-efficacy (*t* = −3.82, *p* < 0.001), tolerance of negative affect (*t* = −3.70, *p* < 0.001), positive support relations (*t* = −2.56, *p* < 0.05), power of control (*t* = −3.05, *p* < 0.01), and spontaneity (*t* = −3.02, *p* < 0.01).

To assess the persistence of these improvements over an extended period (three to four months), including a school vacation, a repeated measures analysis of variance (RMANOVA) text was conducted on the pre-, post-, and follow-up test scores for resilience. This analysis indicated a statistically significant change in resilience over time (*F* = 4.32, *p* < 0.05). Specifically, Tukey’s post-hoc test identified a significant improvement from the pre-test to the post-test but found no significant difference between the pre-test and follow-up scores, as shown in Table 2. Nonetheless, the follow-up test scores (Mean = 119, SD = 1.39) remained higher than the pre-test scores (Mean = 115, SD = 1.55).

For the resilience subscales, statistically significant differences were observed in self-efficacy (*F* = 6.94, *p* < 0.001), tolerance of negative affect (*F* = 6.91, *p* < 0.001), positive support relations (*F* = 3.65, *p* < 0.05), power of control (*F* = 6.35, *p* < 0.001), and spontaneity (*F* = 6.61, *p* < 0.001), as reported in Table 2. Tukey’s post-hoc analysis revealed significant improvement in all resilience subscales between the pre- and post-tests. However, these improvements were not maintained at the follow-up. Nonetheless, the follow-up test means for self-efficacy (M = 28.5, SD = 0.72), tolerance of negative affect (M = 28.2, SD = 0.72), and positive support relations (M = 14.9, SD = 0.38) remained higher than their respective pre-test means (self-efficacy: M = 27.9, SD = 0.65; tolerance of negative affect: M = 27.8, SD = 0.64; positive support relations: M = 14.6, SD = 0.33).

### 4.2. Social–Emotional Competencies

Table 3 presents the pre-test and post-test means and standard deviations for social and emotional competencies, including emotional regulation, social skills, empathy, social tendency, and interpersonal relations. The *t*-tests comparing pre- and post-test scores reveal statistically significant improvements in emotional regulation (*t* = −5.10, *p* < 0.001), social skills (*t* = −3.82, *p* < 0.001), empathy (*t* = −2.40, *p* < 0.05), and social tendency (*t* = −3.43, *p* < 0.001) following the SEE Learning intervention. However, improvements in interpersonal relations were not statistically significant (*t* = −1.81). These results suggest that a short-term intervention program based on SEE Learning effectively enhances elementary students’ emotional regulation, social skills, empathy, and social tendency.

To evaluate the persistence of improvements in social–emotional competencies over an extended period (three to four months), including a school vacation, a repeated measures analysis of variance (RMANOVA) was performed on the pre-, post-, and follow-up test scores for each variable. Statistically significant differences were observed in emotional regulation (*F* = 6.16, *p* < 0.01), social skills (*F* = 6.09, *p* < 0.01), empathy (*F* = 4.26, *p* < 0.05), and social tendency (*F* = 2.95, *p* < 0.05), as reported in Table 4. Tukey’s post-hoc analysis revealed a statistically significant persistence of improvements in social skills between the pre- and post-test (*t* = −3.273, *p* < 0.01), as well as between the pre- and follow-up test (*t* = −2.566, *p* < 0.05). Although statistically significant improvements in emotional regulation, empathy, and social tendency occurred from pre- to post-test, these were not maintained at the follow-up. However, the follow-up test means for emotional regulation (M = 48, SD = 0.49), empathy (M = 31.7, SD = 0.33) and social tendency (M = 31.7, SD = 0.33) remained higher than their respective pre-test means (emotional regulation: M = 47.3, SD = 0.52; empathy: M = 30.8, SD = 0.36; social tendency: M = 30.8, SD = 0.36).

## 5. Discussion

This research project assessed the impacts of a short-term SEE Learning (Social, Emotional, and Ethical Learning) intervention program that focused on resilience and social and emotional competencies and was specifically tailored to elementary students in Daegu, South Korea, the epicenter of South Korea’s initial COVID-19 outbreak. Notably, the participating students, currently in grades 3–4, were in first or second grade—a foundational stage of education—when the pandemic surged in 2020, placing them at the center of the educational disruptions caused by COVID-19. This demographic context is pivotal, as these children have spent crucial developmental years in the shadow of the pandemic, potentially exacerbating their need for effective social and emotional support.

The study’s objective was to examine whether a six-week adaptation of the SEE Learning program could effectively bolster resilience and social–emotional competencies among these young learners. The program, condensed to fit within the school term, focused on key components, such as attention and self-awareness, self-compassion, self-regulation, interpersonal awareness, compassion for others, relationship skills, and appreciating interdependence, that are crucial for navigating and mitigating the psychological impacts of the pandemic as well as for the general improvement of whole-child development.

The results revealed statistically significant improvements in students’ resilience. This finding supports prior research showing that SEL programs can enhance resilience (e.g., [23,24,25]). Resilience refers to the ability to recover healthily from both everyday small traumas, such as difficulties with friends or parents, and significant traumas, such as major life challenges or COVID-19 [21,41]. Therefore, resilience is likely a crucial skill for students who experienced COVID-19, which heightened depression and anxiety and led to relational disconnections. It has been reported that resilience helps people lead more constructive and healthier lives, not only personally but also in interactions with others and within systems [47]. Therefore, the finding that SEE Learning, which integrates trauma-informed education, is an effective tool for addressing these issues among elementary students whose social–emotional development has been limited by the pandemic has important implications for the educational field, which has an urgent need for social–emotional programs. Specifically, teaching 12 lessons over a six-week period was effective. Even after three to four months, although the difference was not statistically significant, average scores remained higher than on the pre-test, indicating that resilience can be effectively taught to students via a short-term intervention. Overall, the results suggest that resilience can be cultivated and improved, thus helping to heal the wounds and challenges incurred by COVID-19 [48]. Further, the study indicates that to maintain these effects over time, it is important to provide tools that parents can use with their children during vacation periods to prompt further development.

The study also revealed that the six-week, 12-session SEE Learning program enhanced social–emotional competencies, which is particularly important for students who have had insufficient practice forming relationships, as well as for students, teachers, and school personnel experiencing emotional difficulties, especially after living through the COVID-19 pandemic. The National Center for Education Statistics in the U.S. Department of Education released data indicating that more than 80% of public schools reported that the pandemic negatively affected students’ socio-emotional development and that these students require additional support in school settings [49]. The current study shows that SEE Learning specifically improves social skills and social tendencies, a result aligning with research conducted by Durlak et al. [22], which found that SEL programs enhance social skills and prosocial behaviors. Moreover, the fact that SEE Learning improved emotional regulation and empathy aligns with the findings of Hong et al. [30]. In other words, just as various studies [28] identified improvements in resilience and social–emotional competencies based on SEL programs, the current results confirm that SEE Learning effectively enhances these skills. These findings indicate that SEE Learning can serve as a crucial tool in facilitating the social–emotional development of elementary students who, due to COVID-19, have been unable to attend school for long periods, instead participating in lessons online. Additionally, SEE Learning can enhance the social–emotional development of students in circumstances when offering a year-long program is not possible and short-term intervention is necessary.

Additionally, the SEE Learning program integrated mindfulness and compassion practices, with the results showing that resilience and social–emotional competencies were effectively enhanced when these elements were incorporated into conventional SEL programs. Mindfulness programs such as Mindfulness-Based Stress Reduction (MBSR) help participants maintain awareness of sensations, feelings, thoughts, and the surrounding environment and sustain their attention. Research indicates that mindfulness programs develop coping skills that help individuals relieve and manage stress or burnout, thereby reducing suffering and enhancing attention, self-awareness, and self-regulation [31,32,33,34,38,39]. Further, higher levels of mindfulness tend to correlate with higher levels of resilience [50], and mindfulness has been found to increase emotional regulation, empathy, and social connectedness [35]. Thus, prior findings and the current results indicate that SEE Learning, which integrates mindfulness, enhances resilience and social–emotional competencies.

Finally, SEE Learning is fundamentally based on compassion, or recognizing and relieving one’s own and others’ suffering while cultivating kindness and positive affection. Despite trauma exposure, higher self-compassion reduces psychological distress [36], and higher compassion for others leads to increased protective and prosocial behaviors, thereby better controlling the stress and anxiety brought on by COVID-19 and fostering resilience and recovery [37,51]. Stevens and Taber [52] taking a neuroscientific perspective, reported that compassion regulates emotions and increases prosocial behavior. It is likely that in this study, SEE Learning, which is founded on compassion, enhanced resilience and social–emotional competencies among students who had returned to face-to-face classes after the COVID-19 lockdowns, in part due to the cultivation of compassion. Therefore, SEE Learning—an SEL 2.0 that integrates mindfulness and compassion—can likely enhance resilience and social–emotional competencies, thereby facilitating school adjustment and fostering a healthy and satisfying life, not only for students affected by the pandemic but also as part of the general curriculum of school systems [53,54,55].

### Limitations & Future Studies

No study is without limitations. First, we conducted 12 learning sessions over six weeks based on the time constraints of the school curriculum. Extending the duration, if time permits, might allow for more comprehensive practice and a re-evaluation of the SEE Learning curriculum’s effectiveness. Thus, future studies should apply the full SEE Learning curriculum over a year-long period to explore its effectiveness. Additionally, researchers should examine whether creating and implementing a booster program enhances the sustainability of the program’s effects.

Further, while this study demonstrated the positive impacts of the SEE Learning curriculum on elementary students’ resilience, emotional regulation, social skills, empathy, and social tendencies, it did not examine the relationships between these variables. Additionally, although SEE Learning incorporates mindfulness and compassion, this study did not assess the mindfulness and compassion levels of participating students because there are no measurement tools suitable for elementary students. Therefore, future research should investigate additional variables and their interrelations to better understand the mechanisms underlying the effects of the SEE Learning curriculum.

Lastly, this study was conducted in public schools in South Korea and was unable to perform a comparative analysis due to the absence of a control group. At the outset of the study, the COVID-19 pandemic had not yet concluded, resulting in stringent standards for curriculum operation and external programs. Additionally, given that the target group comprised lower-grade students, both the Office of Education and the school administration did not permit the inclusion of a control group. Therefore, it is important for future studies to include a control group to assess the effectiveness of the program more accurately.

## 6. Conclusions

In conclusion, the SEE Learning short-term intervention program adapted to the context of COVID-19’s educational disruptions demonstrates substantial promise in enhancing resilience and social–emotional competencies among elementary students. Specifically, the effects of SEE Learning were assessed in Daegu, where students experienced direct consequences from the first outbreak of COVID-19; understanding the pandemic’s specific impacts on students who were in first and second grade at the pandemic’s onset can provide deeper insight into the program’s efficacy for students under extreme stress. The findings highlight the importance of broader implementation and further exploration of SEE Learning in similar educational settings, especially in areas directly impacted by the pandemic. More generally, this study underscores the urgent need for educational strategies that extend beyond traditional learning paradigms, embracing an integrative approach that includes the mental, social, and emotional well-being of students as central components of their educational journey. Further research will be vital in deepening scholars’ understanding and refinement of social and emotional interventions.

## Figures and Tables

**Table 1 behavsci-14-00458-t001:** Pre- and post-test means and standard deviations for resilience.

Variable	n	Pre-Test	Post-Test	*t*-Value
		M (SD)	M (SD)	
Resilience—Total	292	113 (22.4)	120 (19.8)	−4.25 ***
Resilience—Self-Efficacy	305	30.05 (6.11)	31.84 (5.48)	−3.82 ***
Resilience—Tolerance of Negative Affect	300	29.87 (6.00)	31.62 (5.62)	−3.70 ***
Resilience—Positive Support Relations	306	15.61 (3.00)	16.21 (2.86)	−2.56 *
Resilience—Power of Control	302	23.64 (4.38)	24.67 (3.90)	−3.05 **
Resilience—Spontaneity	305	15.32 (3.31)	16.10 (3.01)	−3.02 **

*Note*: *** *p* < 0.001; ** *p* < 0.01; * *p* < 0.05.; n—number of participants.

**Table 2 behavsci-14-00458-t002:** Pre- and post-test means and standard deviations for resilience, including follow-up tests and post-hoc tests.

Variable	n	Pre-Test	Post-Test	Follow-Up	*F*-Value (*p*-Value)	Pre-Post	Pre-Follow-Up	Post-Follow-Up
		M (SD)	M (SD)	M (SD)		*t*-Value	*t*-Value	*t*-Value
Resilience (total)	182	115 (1.55)	121 (1.50)	119 (1.39)	4.32 *	−2.783 *	−1.966	0.939
Self-Efficacy	217	27.9 (0.65)	30.9 (0.53)	28.5 (0.72)	6.94 ***	−3.813 ***	−0.576	2.863 *
Tolerance of Negative Affect	218	27.8 (0.64)	30.8 (0.53)	28.2 (0.72)	6.91 ***	−3.729 ***	−0.421	2.904 *
Positive Support Relations	218	14.6 (0.33)	15.8 (0.27)	14.9 (0.38)	3.65 *	−2.966 ***	−0.674	1.834
Power of Control	218	22.1 (0.49)	24.1 (0.39)	22.1 (0.55)	6.35 ***	−3.386 ***	−0.013	3.035 ***
Spontaneity	218	14.2 (0.34)	15.7 (0.29)	14.2 (0.38)	6.61 ***	−3.478 ***	−0.056	3.085 ***

*Note*: *** *p* < 0.001; * *p* < 0.05; n—number of participants; M (SD), mean ± standard deviation; repeated measures analysis of variance was conducted; post-hoc, Tukey’s test.

**Table 3 behavsci-14-00458-t003:** Pre- and post-test means and standard deviations for social–emotional competencies.

Variable	n	Pre-Test	Post-Test	*t*-Value (*p*-Value)
		M (SD)	M (SD)	
Emotional Regulation	299	46.78 (6.73)	49.80 (7.81)	−5.10 ***
Social Skills	306	46.42 (7.43)	48.86 (8.36)	−3.82 ***
Empathy	313	23.42 (3.70)	24.17 (4.04)	−2.40 *
Social Tendency	308	30.42 (4.75)	31.73 (4.79)	−3.43 ***
Interpersonal Relations	315	28.80 (3.92)	29.39 (4.39)	−1.81

*Note*: *** *p* < 0.001; * *p* < 0.05; n—number of participants; M (SD), mean ± standard deviation.

**Table 4 behavsci-14-00458-t004:** Pre- and post-test means and standard deviations for social–emotional competencies, including follow-up tests and post-hoc tests.

Variable	n	Pre-Test	Post-Test	Follow-Up	*F*-Value (*p*-Value)	Pre-Post	Pre-Follow-Up	Post-Follow-Up
		M (SD)	M (SD)	M (SD)		*t*-Value	*t*-Value	*t*-Value
ER	184	47.3 (0.52)	49.9 (0.61)	48 (0.49)	6.16 **	−3.305 **	−0.867	2.499 *
SS	184	46.6 (0.56)	49.4 (0.64)	48.6 (0.54)	6.09 **	−3.273 **	−2.566 *	0.939
Empathy	184	23.4 (0.28)	24.5 (0.31)	23.7 (0.29)	4.26 *	−2.731 *	−0.562	2.238
ST	184	30.8 (0.36)	32.1 (0.38)	31.7 (0.33)	2.95 *	−2.382 *	−1.633	0.687
IR	184	28.8 (0.31)	29.6 (0.34)	29.4 (0.30)	2.03	−1.974	−1.512	0.406

*Note*: ** *p* < 0.01; * *p* < 0.05; n—number of participants; ER—emotional regulation; SS—social skills; ST—social tendency; IR—interpersonal relations.; M (SD), mean ± standard deviation; repeated measures analysis of variance was conducted; post-hoc, Tukey’s test.

## Data Availability

Subsets of data are available from the corresponding author upon reasonable request.

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
