# Peer review of "Building Resilience and Social–Emotional Competencies in Elementary School Students through a Short-Term Intervention Program Based on the SEE Learning Curriculum"

_behavsci, 2024, doi:10.3390/bs14060458_

Round 1

Reviewer 1 Report

Comments and Suggestions for Authors

Thank you for the opportunity to review the manuscript entitled Building Resilience and Social-Emotional Competencies in Elementary School Students Through a Short-Term Intervention Program based on the SEE Learning Curriculum for Behavioral Sciences

The study described addresses a timely and important issue. The following are suggestions to improve the manuscript: 

*The topics included in the 12-session program were listed. The authors state that they reconfigured the program to incorporate 12 learning experiences over 6 weeks. However, it's not clear how the 12 sessions that were included in the program were chosen. I'd like to hear more about that decision/selection process. How were the original topics narrowed down? How many were in the original curriculum? 

*I would like to hear more about what occurred in the training sessions. What was the structure/format of the lessons? What kinds of activities were included? Where the lessons teacher directed, cooperative, etc? What did students (& teachers) do during the lessons?

Author Response

Dear Reviewer,

Thank you for your feedback. We have sincerely accepted your feedback and conducted a revision based on your suggestions. We appreciate you.

Please check out the following revisions:

Comments: The topics included in the 12-session program were listed. The authors state that they reconfigured the program to incorporate 12 learning experiences over 6 weeks. However, it's not clear how the 12 sessions that were included in the program were chosen. I'd like to hear more about that decision/selection process. How were the original topics narrowed down? How many were in the original curriculum? 

Authors' notes: 

Page 5, it has been described about original curriculum contents.

"Specifically, SEE Learning curriculum includes 43 learning experiences spread across 9 chapters. These chapters cover topics such as creating a compassionate classroom, building resilience, strengthening attention and self-awareness, navigating emotions, learning about and from one another, compassion for self and others, we are all in this together, and building a better world. For examples, students learn, experience, and practice kindness by reflecting on the acts of kindness they have received through drawing activities. They explore diverse resilience strategies using ‘Help! Now!’ strategies and learn about the importance of attention and how it benefits them by cultivating mindfulness, awareness, and heedfulness. The curriculum also addresses understanding the needs behind various emotions and mechanism to manage them, handling impulsive behaviors, and gaining insights into others’ lives through mindful interviews. Students recognize commonalities and differences, cultivate kindness and compassion for themselves and others, and understand systems within their classrooms, schools, and neighborhoods. They also appreciate interdependence through activities that help think about people related to their favorite food, achievements, and other vital aspects, and apply these insights within their communities."

For the selection process, please refer to the following:

It was necessary to modify the curriculum to fit into a short six-week period in public schools to enhance resilience and social and emotional competencies. Thus, 12 sessions were selected that were enhancing resilience and social emotional skills. We reconfigured the program to incorporate 12 learning experiences over six weeks, focusing on aspects such as attention and self-awareness, self-compassion, self-regulation, interpersonal awareness, compassion for others, relationship skills, and appreciating interdependence.

Comments: I would like to hear more about what occurred in the training sessions. What was the structure/format of the lessons? What kinds of activities were included? Where the lessons teacher directed, cooperative, etc? What did students (& teachers) do during the lessons?

Authors' notes: 

For teacher training, six experienced instructors who had completed official teacher trainings for the 43 SEE Learning programs, learned, and practiced each learning activity of the SEE Learning curriculum and had prior experience educating and counseling elementary students, led the program in schools.

We have added what teachers and students did during the lessons.

"For the students, the six-week, 12-session program based on SEE Learning focused on improving resilience and social and emotional competencies among elementary students. In the first week, students engaged in activities including self-introduction with an “in and out” activity, exploring kindness, and remembering kindness received. In the second week, they established classroom rules and practiced kindness. In the third week, they explored authentic kindness, conducted interdependence activities, and considered kindness connected to their achievements. In the fourth week, they contemplated body sensations, discovered new things through attention activities, practiced “Help! Now!” strategies and learned how to resource. In the fifth week, they created a happy treasure box, practiced grounding techniques, and learned how to stay in the resilient zone. In the final week, they delved deeply into the resilient zone, learned about the nervous system, and explored nurturing mindfulness, awareness, and compassion together, using the case of understanding how to walk a dog. This short-term intervention program allowed students to discuss and actively engage in activities designed to improve resilience and social-emotional skills."

"Instructors facilitated these activities, and students actively engaged in each lesson by moving, drawing, acting, discussing the subjects."

Thank you again for your comments!

Reviewer 2 Report

Comments and Suggestions for Authors

The article deals with an important topic, the development of social and emotional skills in primary school pupils. The article is clearly written, and the content is structured and coherent. The research seems innovative given the paucity of studies evaluating the SEE Learning program for primary school pupils.

The introduction is well argued, and the problem is clear. It is therefore easy to grasp the reasons that led to the research question. On p. 2, it would have been interesting to cite one or two references to support the sentence "Various other SEL programs, combined...".

The theoretical section is also well developed and well referenced. However, on p. 5, it would have been interesting to go further in the presentation of the SEE Learning program by illustrating it with concrete activities (e.g., describing a compassion and kindness activity, or an ethical discernment activity). Furthermore, in point 2.3 (p. 5) on the research question, it is noted that "few empirical studies have been conducted on the outcomes..." of SEE learning programs. Even if there are few, it would have been interesting to present them and show what your research adds or differs.

The method section does not specify how the measurement instruments were administered to the students (by whom? how? time required? support provided? etc.). The two instruments used have a total of 70 items, which can be difficult to read for pupils of this age and, above all, can be very time-consuming. This may also raise ethical issues (duration and conditions of the test) for such young children, ethical concerns that are not addressed. It would therefore be good to have details of how the test was administered. We also don't know whether this was problematic for certain pupils, for example those with special educational needs. More details are needed in terms of the data collection procedure. A more precise description of the participants would also be welcome (special educational needs? age of pupils? etc.).

The methods section should also better explain the decision to carry out a SEE Learning program over only six weeks. Even though this is mentioned in the limitations, the reasons for it remain difficult for an external reader to understand. Why was a full-year program not possible? What were the limitations? What other arguments are there to support such a short program? These points could also be better discussed in the limitations section.

Still in the methods section or in the 'limitations' section, the reasons why the school's curriculum does not allow for a control group should be better explained. The authors gloss over this aspect too quickly, even though it is a major limitation.

The results are unfortunate in that they remain very general and do not indicate whether gender differences were observed, or whether differences were observed according to school level (grades 3 or 4). This could add nuance and enrich the discussion, as well as the prospects for future research. In addition, the authors do not specify whether there were any differences between pupils at the outset (e.g. between boys and girls or by class) or why this is not investigated.

As far as the discussion is concerned, we know that training pupils without training teachers, i.e. not adopting a 'whole school approach', can have its limits. The authors do not address this aspect sufficiently.

In terms of form, table 4 contains lines that should be deleted. On p. 4, the Slavich, Roos, & Zaki reference should, according to APA 7th standards, be presented directly as "Slavich et al., 2022". Please check with the journal standards.

In the same parenthesis, again at the bottom of page 4, there is an unnecessary semicolon before the final parenthesis ("... Mettler et al., 2023 ;)" instead of "... Mettler et al., 2023)).

Author Response

Dear Reviewer,

Thank you for your feedback.

We have sincerely accepted your feedback and conducted a revision based on your suggestions.

We deeply appreciate you.  

Please see the attachment for our revision. 

Reviewer 3 Report

Comments and Suggestions for Authors

The evaluated article presents an intervention study in which a program aimed at improving resilience and emotional competencies in elementary school students is analyzed. The topic of the manuscript is relevant and of great value for the educational field. The sample is large and the analyzes carried out are rigorous. Furthermore, the results obtained point in a positive direction and they are a significant contribution to the field of emotional education.

For all these reasons, I consider the article to be of high quality and recommend its publication in the journal. It is only detected that the objectives and hypotheses of the research are not clearly described, which is why their explicit inclusion is requested. It can be done at the end of the introduction or in a separate section.

Author Response

Dear Reviewer,

Thank you very much for your feedback.

We have sincerely accepted your feedback and conducted a revision based on your suggestions.

We deeply appreciate you.

Please check out the following revisions:

Comments: The evaluated article presents an intervention study in which a program aimed at improving resilience and emotional competencies in elementary school students is analyzed. The topic of the manuscript is relevant and of great value for the educational field. The sample is large and the analyzes carried out are rigorous. Furthermore, the results obtained point in a positive direction and they are a significant contribution to the field of emotional education. For all these reasons, I consider the article to be of high quality and recommend its publication in the journal.

Authors' notes: Thank you very much for your comments!!

Comments: It is only detected that the objectives and hypotheses of the research are not clearly described, which is why their explicit inclusion is requested. It can be done at the end of the introduction or in a separate section.

Authors' notes: We have added the purpose of the study on page 3. 

“This study aims to develop and implement a short-term intervention program to apply the SEE Learning program in a post-COVID-19 face-to-face setting, and to explore its effectiveness. The necessity of developing a short-term intervention arises in situations such as those brought about by COVID-19, where normal curriculum progress is challenging, when the entire curriculum is too lengthy to be incorporated into the public standardized educational system, or when there is a lack of understanding of the curriculum. Therefore, in this post-COVID-19 situation and under circumstances where a year-long curriculum cannot be implemented, we aimed to develop and apply a short-term intervention program of the SEE Learning program and investigate its effectiveness.

To achieve the purpose of the study, we implemented a short-term intervention based on the SEE Learning program, which integrates resilience, attention training, and compassion, in Daegu area elementary schools where face-to-face classes had resumed following the initial COVID-19 outbreaks. We then explored changes in students' social and emotional development, including resilience, emotional regulation, social skills, and empathy. In this article, we outline the SEE Learning program utilized in the study and discuss the relevance and development of social and emotional learning in the context of COVID-19. Additionally, we examine the emotional and relational shifts students experienced after program implementation and the broader implications of these shifts.”

Thank you again for your comments.